# PAC-Bayesian AUC classification and scoring

**James Ridgway**[*]
CREST and CEREMADE University Dauphine
james.ridgway@ensae.fr

**Pierre Alquier**
CREST (ENSAE)
pierre.alquier@ucd.ie

**Nicolas Chopin**
CREST (ENSAE) and HEC Paris
nicolas.chopin@ensae.fr

**Feng Liang**
University of Illinois at Urbana-Champaign
liangf@illinois.edu

## Abstract

We develop a scoring and classification procedure based on the PAC-Bayesian approach and the AUC (Area Under Curve) criterion. We focus initially on the class of linear score functions. We derive PAC-Bayesian non-asymptotic bounds for two types of prior for the score parameters: a Gaussian prior, and a spike-and-slab prior; the latter makes it possible to perform feature selection. One important advantage of our approach is that it is amenable to powerful Bayesian computational tools. We derive in particular a Sequential Monte Carlo algorithm, as an efficient method which may be used as a gold standard, and an Expectation-Propagation algorithm, as a much faster but approximate method. We also extend our method to a class of non-linear score functions, essentially leading to a nonparametric procedure, by considering a Gaussian process prior.

## 1 Introduction

Bipartite ranking (scoring) amounts to rank (score) data from binary labels. An important problem in its own right, bipartite ranking is also an elegant way to formalise classification: once a score function has been estimated from the data, classification reduces to chooses a particular threshold, which determine to which class is assigned each data-point, according to whether its score is above or below that threshold. It is convenient to choose that threshold only once the score has been estimated, so as to get finer control of the false negative and false positive rates; this is easily achieved by plotting the ROC (Receiver operating characteristic) curve.

A standard optimality criterion for scoring is AUC (Area Under Curve), which measures the area under the ROC curve. AUC is appealing for at least two reasons. First, maximising AUC is equivalent to minimising the $L_1$ distance between the estimated score and the optimal score. Second, under mild conditions, Cortes and Mohri [2003] show that AUC for a score $s$ equals the probability that $s(X^-) < s(X^+)$ for $X^-$ (resp. $X^+$) a random draw from the negative (resp. positive class). Yan et al. [2003] observed AUC-based classification handles much better skewed classes (say the positive class is much larger than the other) than standard classifiers, because it enforces a small score for all members of the negative class (again assuming the negative class is the smaller one).

One practical issue with AUC maximisation is that the empirical version of AUC is not a continuous function. One way to address this problem is to "convexify" this function, and study the properties of so-obtained estimators [Clémençon et al., 2008a]. We follow instead the PAC-Bayesian approach in this paper, which consists of using a random estimator sampled from a pseudo-posterior distribution that penalises exponentially the (in our case) AUC risk. It is well known [see e.g. the monograph of Catoni, 2007] that the PAC-Bayesian approach comes with a set of powerful technical tools to

---

[*]http://www.crest.fr/pagesperso.php?user=3328

establish non-asymptotic bounds; the first part of the paper derive such bounds. A second advantage however of this approach, as we show in the second part of the paper, is that it is amenable to powerful Bayesian computational tools, such as Sequential Monte Carlo and Expectation Propagation.

## 2  Theoretical bounds from the PAC-Bayesian Approach

### 2.1  Notations

The data $\mathcal{D}$ consist in the realisation of $n$ IID (independent and identically distributed) pairs $(X_i, Y_i)$ with distribution $P$, and taking values in $\mathbb{R}^d \times \{-1, 1\}$. Let $n_+ = \sum_{i=1}^{n} \mathbb{1}\{Y_i = +1\}$, $n_- = n - n_+$. For a score function $s : \mathbb{R}^d \to \mathbb{R}$, the AUC risk and its empirical counter-part may be defined as:

$$R(s) = \mathbb{P}_{(X,Y),(X',Y') \sim P} \left[ \{s(X) - s(X')\}(Y - Y') < 0 \right],$$

$$R_n(s) = \frac{1}{n(n-1)} \sum_{i \neq j} \mathbb{1} \left[ \{s(X_i) - s(X_j)\}(Y_i - Y_j) < 0 \right].$$

Let $\sigma(x) = \mathbb{E}(Y|X = x)$, $\bar{R} = R(\sigma)$ and $\bar{R}_n = R_n(\sigma)$. It is well known that $\sigma$ is the score that minimise $R(s)$, i.e. $R(s) \geq \bar{R} = R(\sigma)$ for any score $s$.

The results of this section apply to the class of linear scores, $s_\theta(x) = \langle \theta, x \rangle$, where $\langle \theta, x \rangle = \theta^T x$ denotes the inner product. Abusing notations, let $R(\theta) = R(s_\theta)$, $R_n(\theta) = R_n(s_\theta)$, and, for a given prior density $\pi_\xi(\theta)$ that may depend on some hyperparameter $\xi \in \Xi$, define the Gibbs posterior density (or pseudo-posterior) as

$$\pi_{\xi, \gamma}(\theta | \mathcal{D}) := \frac{\pi_\xi(\theta) \exp\{-\gamma R_n(\theta)\}}{Z_{\xi, \gamma}(\mathcal{D})}, \quad Z_{\xi, \gamma}(\mathcal{D}) = \int_{R^d} \pi_\xi(\tilde{\theta}) \exp\left\{-\gamma R_n(\tilde{\theta})\right\} \mathrm{d}\tilde{\theta}$$

for $\gamma > 0$. Both the prior and posterior densities are defined with respect to the Lebesgue measure over $\mathbb{R}^d$.

### 2.2  Assumptions and general results

Our general results require the following assumptions.

**Definition 2.1**  *We say that Assumption* **Dens**$(c)$ *is satisfied for $c > 0$ if*

$$\mathbb{P}(\langle X_1 - X_2, \theta \rangle \geq 0, \langle X_1 - X_2, \theta' \rangle \leq 0) \leq c \|\theta - \theta'\|$$

*for any $\theta$ and $\theta' \in \mathbb{R}^d$ such that $\|\theta\| = \|\theta'\| = 1$.*

This is a mild Assumption, which holds for instance as soon as $(X_1 - X_2)/\|X_1 - X_2\|$ admits a bounded probability density; see the supplement.

**Definition 2.2 (Mammen & Tsybakov margin assumption)**  *We say that Assumption* **MA**$(\kappa, C)$ *is satisfied for $\kappa \in [1, +\infty]$ and $C \geq 1$ if*

$$\mathbb{E}\left[(q_{1,2}^\theta)^2\right] \leq C \left[R(\theta) - \overline{R}\right]^{\frac{1}{\kappa}}$$

*where $q_{i,j}^\theta = \mathbb{1}\{\langle \theta, X_i - X_j \rangle (Y_i - Y_j) < 0\} - \mathbb{1}\{[\sigma(X_i) - \sigma(X_j)](Y_i - Y_j) < 0\} - R(\theta) + \overline{R}$.*

This assumption was introduced for classification by Mammen and Tsybakov [1999], and used for ranking by Clémençon et al. [2008b] and Robbiano [2013] (see also a nice discussion in Lecué [2007]). The larger $\kappa$, the less restrictive **MA**$(\kappa, C)$. In fact, **MA**$(\infty, C)$ is always satisfied for $C = 4$. For a noiseless classification task (i.e. $\sigma(X_i)Y_i \geq 0$ almost surely), $\overline{R} = 0$,

$$\mathbb{E}((q_{1,2}^\theta)^2) = \mathrm{Var}(q_{1,2}^\theta) = \mathbb{E}[\mathbb{1}\{\langle \theta, X_1 - X_2 \rangle (Y_i - Y_j) < 0\}] = R(\theta) - \overline{R}$$

and **MA**$(1, 1)$ holds. More generally, **MA**$(1, C)$ is satisfied as soon as the noise is small; see the discussion in Robiano 2013 (Proposition 5 p. 1256) for a formal statement. From now, we focus on either **MA**$(1, C)$ or **MA**$(\infty, C)$, $C \geq 1$. It is possible to prove convergence under **MA**$(\kappa, 1)$

for a general $\kappa \geq 1$, but at the price of complications regarding the choice of $\gamma$; see Catoni [2007], Alquier [2008] and Robbiano [2013].

We use the classical PAC-Bayesian methodology initiated by McAllester [1998]; Shawe-Taylor and Williamson [1997] (see Alquier [2008]; Catoni [2007] for a complete survey and more recent advances) to get the following results. Proof of these and forthcoming results may be found in the supplement. Let $\mathcal{K}(\rho, \pi)$ denotes the Kullback-Liebler divergence, $\mathcal{K}(\rho, \pi) = \int \rho(\mathrm{d}\theta) \log\{\frac{\mathrm{d}\rho}{\mathrm{d}\pi}(\theta)\}$ if $\rho << \pi$, $\infty$ otherwise, and denote $\mathcal{M}_+^1$ the set of probability distributions $\rho(\mathrm{d}\theta)$.

**Lemma 2.1** *Assume that* $\mathbf{MA}(1, C)$ *holds with* $C \geq 1$. *For any fixed* $\gamma$ *with* $0 < \gamma \leq (n-1)/(8C)$, *for any* $\varepsilon > 0$, *with probability at least* $1 - \varepsilon$ *on the drawing of the data* $\mathcal{D}$,

$$\int R(\theta)\pi_{\xi,\gamma}(\theta|\mathcal{D})\mathrm{d}\theta - \overline{R} \leq 2 \inf_{\rho \in \mathcal{M}_+^1} \left\{ \int R(\theta)\rho(\mathrm{d}\theta) - \overline{R} + 2\frac{\mathcal{K}(\rho, \pi) + \log\left(\frac{4}{\varepsilon}\right)}{\gamma} \right\}.$$

**Lemma 2.2** *Assume* $\mathbf{MA}(\infty, C)$ *with* $C \geq 1$. *For any fixed* $\gamma$ *with* $0 < \gamma \leq (n-1)/8$, *for any* $\epsilon > 0$ *with probability* $1 - \epsilon$ *on the drawing of* $\mathcal{D}$,

$$\int R(\theta)\pi_{\xi,\gamma}(\theta|\mathcal{D})\mathrm{d}\theta - \bar{R} \leq \inf_{\rho \in \mathcal{M}_+^1} \left\{ \int R(\theta)\rho(d\theta) - \bar{R} + 2\frac{\mathcal{K}(\rho, \pi) + \log\frac{2}{\epsilon}}{\gamma} \right\} + \frac{16\gamma}{n-1}.$$

Both lemmas bound the expected risk excess, for a random estimator of $\theta$ generated from $\pi_{\xi,\gamma}(\theta|\mathcal{D})$.

## 2.3 Independent Gaussian Prior

We now specialise these results to the prior density $\pi_\xi(\theta) = \prod_{i=1}^d \varphi(\theta_i; 0, \vartheta)$, i.e. a product of independent Gaussian distributions $N(0, \vartheta)$; $\xi = \vartheta$ in this case.

**Theorem 2.3** *Assume* $\mathbf{MA}(1, C)$, $C \geq 1$, $\mathbf{Dens}(c)$, $c > 0$, *and take* $\vartheta = \frac{2}{d}(1 + \frac{1}{n^2 d})$, $\gamma = (n-1)/8C$, *then there exists a constant* $\alpha = \alpha(c, C, d)$ *such that for any* $\epsilon > 0$, *with probability* $1 - \epsilon$,

$$\int R(\theta)\pi_\gamma(\theta|\mathcal{D})\mathrm{d}\theta - \bar{R} \leq 2\inf_{\theta_0} \left\{ R(\theta_0) - \bar{R} \right\} + \alpha \frac{d\log(n) + \log\frac{4}{\epsilon}}{n-1}.$$

**Theorem 2.4** *Assume* $\mathbf{MA}(\infty, C)$, $C \geq 1$, $\mathbf{Dens}(c)$ $c > 0$, *and take* $\vartheta = \frac{2}{d}(1 + \frac{1}{n^2 d})$, $\gamma = C\sqrt{dn\log(n)}$, *there exists a constant* $\alpha = \alpha(c, C, d)$ *such that for any* $\epsilon > 0$, *with probability* $1 - \epsilon$,

$$\int R(\theta)\pi_\gamma(\theta|\mathcal{D})\mathrm{d}\theta - \bar{R} \leq \inf_{\theta_0} \left\{ R(\theta_0) - \bar{R} \right\} + \alpha \frac{\sqrt{d\log(n)} + \log\frac{2}{\epsilon}}{\sqrt{n}}.$$

The proof of these results is provided in the supplementary material. It is known that, under $\mathbf{MA}(\kappa, C)$, the rate $(d/n)^{\frac{\kappa}{2\kappa-1}}$ is minimax-optimal for classification problems, see Lecué [2007]. Following Robbiano [2013] we conjecturate that this rate is also optimal for ranking problems.

## 2.4 Spike and slab prior for feature selection

The independent Gaussian prior considered in the previous section is a natural choice, but it does not accommodate sparsity, that is, the possibility that only a small subset of the components of $X_i$ actually determine the membership to either class. For sparse scenarios, one may use the spike and slab prior of Mitchell and Beauchamp [1988], George and McCulloch [1993],

$$\pi_\xi(\theta) = \prod_{i=1}^d \left[ p\varphi(\theta_i; 0, v_1) + (1-p)\varphi(\theta_i; 0, v_0) \right]$$

with $\xi = (p, v_0, v_1) \in [0, 1] \times (\mathbb{R}^+)^2$, and $v_0 \ll v_1$, for which we obtain the following result. Note $\|\theta\|_0$ is the number of non-zero coordinates for $\theta \in \mathbb{R}^d$.

**Theorem 2.5** *Assume* $\mathbf{MA}(1, C)$ *holds with* $C \geq 1$, $\mathbf{Dens}(c)$ *holds with* $c > 0$, *and take* $p = 1 - \exp(-1/d)$, $v_0 \leq 1/(2nd\log(d))$, *and* $\gamma = (n-1)/(8C)$. *Then there is a constant* $\alpha = \alpha(C, v_1, c)$ *such that for any* $\varepsilon > 0$, *with probability at least* $1 - \varepsilon$ *on the drawing of the data* $\mathcal{D}$,

$$\int R(\theta)\pi_\gamma(\mathrm{d}\theta|\mathcal{D}) - \overline{R} \leq 2\inf_{\theta_0}\left\{ R(\theta_0) - \overline{R} + \alpha\frac{\|\theta_0\|_0 \log(nd) + \log\left(\frac{4}{\varepsilon}\right)}{2(n-1)} \right\}.$$

Compared to Theorem 2.3, the bound above increases logarithmically rather than linearly in $d$, and depends explicitly on $\|\theta\|_0$, the sparsity of $\theta$. This suggests that the spike and slab prior should lead to better performance than the Gaussian prior in sparse scenarios. The rate $\|\theta\|_0 \log(d)/n$ is the same as the one obtained in sparse regression, see e.g. Bühlmann and van de Geer [2011].

Finally, note that if $v_0 \to 0$, we recover the more standard prior which assigns a point mass at zero for every component. However this leads to a pseudo-posterior which is a mixture of $2^d$ components that mix Dirac masses and continuous distributions, and thus which is more difficult to approximate (although see the related remark in Section 3.4 for Expectation-Propagation).

## 3 Practical implementation of the PAC-Bayesian approach

### 3.1 Choice of hyper-parameters

Theorems 2.3, 2.4, and 2.5 propose specific values for hyper-parameters $\gamma$ and $\xi$, but these values depend on some unknown constant $C$. Two data-driven ways to choose $\gamma$ and $\xi$ are (i) cross-validation (which we will use for $\gamma$), and (ii) (pseudo-)evidence maximisation (which we will use for $\xi$).

The latter may be justified from intermediate results of our proofs in the supplement, which provide an empirical bound on the expected risk:

$$\int R(\theta)\pi_{\xi,\gamma}(\theta|\mathcal{D})\mathrm{d}\theta - \bar{R} \leq \Psi_{\gamma,n}\inf_{\rho\in\mathcal{M}_+^1}\left(\int R_n(\theta)\rho(d\theta) - \bar{R}_n + 2\frac{\mathcal{K}(\rho,\pi) + \log\frac{2}{\epsilon}}{\gamma}\right)$$

with $\Psi_{\gamma,n} \leq 2$. The right-hand side is minimised at $\rho(\mathrm{d}\theta) = \pi_{\xi,\gamma}(\theta|\mathcal{D})\mathrm{d}\theta$, and the so-obtained bound is $-\Psi_{\gamma,n}\log(Z_{\xi,\gamma}(\mathcal{D}))/\gamma$ plus constants. Minimising the upper bound with respect to hyperparameter $\xi$ is therefore equivalent to maximising $\log Z_{\xi,\gamma}(\mathcal{D})$ with respect to $\xi$. This is of course akin to the empirical Bayes approach that is commonly used in probabilistic machine learning. Regarding $\gamma$ the minimization is more cumbersome because the dependence with the $\log(2/\epsilon)$ term and $\Psi_{n,\gamma}$, which is why we recommend cross-validation instead.

It seems noteworthy that, beside Alquier and Biau [2013], very few papers discuss the practical implementation of PAC-Bayes, beyond some brief mention of MCMC (Markov chain Monte Carlo). However, estimating the normalising constant of a target density simulated with MCMC is notoriously difficult. In addition, even if one decides to fix the hyperparameters to some arbitrary value, MCMC may become slow and difficult to calibrate if the dimension of the sampling space becomes large. This is particularly true if the target does not (as in our case) have some specific structure that make it possible to implement Gibbs sampling. The two next sections discuss two efficient approaches that make it possible to approximate both the pseudo-posterior $\pi_{\xi,\gamma}(\theta|\mathcal{D})$ and its normalising constant, and also to perform cross-validation with little overhead.

### 3.2 Sequential Monte Carlo

Given the particular structure of the pseudo-posterior $\pi_{\xi,\gamma}(\theta|\mathcal{D})$, a natural approach to simulate from $\pi_{\xi,\gamma}(\theta|\mathcal{D})$ is to use tempering SMC [Sequential Monte Carlo Del Moral et al., 2006] that is, define a certain sequence $\gamma_0 = 0 < \gamma_1 < \ldots < \gamma_T$, start by sampling from the prior $\pi_\xi(\theta)$, then applies successive importance sampling steps, from $\pi_{\xi,\gamma_{t-1}}(\theta|\mathcal{D})$ to $\pi_{\xi,\gamma_t}(\theta|\mathcal{D})$, leading to importance weights proportional to:

$$\frac{\pi_{\xi,\gamma_t}(\theta|\mathcal{D})}{\pi_{\xi,\gamma_{t-1}}(\theta|\mathcal{D})} \propto \exp\left\{ -(\gamma_t - \gamma_{t-1})R_n(\theta) \right\}.$$

When the importance weights become too skewed, one rejuvenates the particles through a resampling step (draw particles randomly with replacement, with probability proportional to the weights) and a move step (move particles according to a certain MCMC kernel).

One big advantage of SMC is that it is very easy to make it fully adaptive. For the choice of the successive $\gamma_t$, we follow Jasra et al. [2007] in solving numerically (1) in order to impose that the Effective sample size has a fixed value. This ensures that the degeneracy of the weights always remain under a certain threshold. For the MCMC kernel, we use a Gaussian random walk Metropolis step, calibrated on the covariance matrix of the resampled particles. See Algorithm 1 for a summary.

---

**Algorithm 1** Tempering SMC

---

**Input** $N$ (number of particles), $\tau \in (0,1)$ (ESS threshold), $\kappa > 0$ (random walk tuning parameter)

**Init.** Sample $\theta_0^i \sim \pi_\xi(\theta)$ for $i = 1$ to $N$, set $t \leftarrow 1$, $\gamma_0 = 0$, $Z_0 = 1$.

**Loop** **a.** Solve in $\gamma_t$ the equation

$$\frac{\{\sum_{i=1}^N w_t(\theta_{t-1}^i)\}^2}{\sum_{i=1}^N \{w_t(\theta_{t-1}^i))^2\}} = \tau N, \quad w_t(\theta) = \exp[-(\gamma_t - \gamma_{t-1})R_n(\theta)] \qquad (1)$$

using bisection search. If $\gamma_t \geq \gamma_T$, set $Z_T = Z_{t-1} \times \left\{ \frac{1}{N} \sum_{i=1}^N w_t(\theta_{t-1}^i) \right\}$, and stop.

**b.** Resample: for $i = 1$ to $N$, draw $A_t^i$ in $1, \ldots, N$ so that $\mathbb{P}(A_t^i = j) = w_t(\theta_{t-1}^j)/\sum_{k=1}^N w_t(\theta_{t-1}^k)$; see Algorithm 1 in the supplement.

**c.** Sample $\theta_t^i \sim M_t(\theta_{t-1}^{A_t^i}, \mathrm{d}\theta)$ for $i = 1$ to $N$ where $M_t$ is a MCMC kernel that leaves invariant $\pi_t$; see Algorithm 2 in the supplement for an instance of such a MCMC kernel, which takes as an input $S = \kappa\hat{\Sigma}$, where $\hat{\Sigma}$ is the covariance matrix of the $\theta_{t-1}^{A_t^i}$.

**d.** Set $Z_t = Z_{t-1} \times \left\{ \frac{1}{N} \sum_{i=1}^N w_t(\theta_{t-1}^i) \right\}$.

---

In our context, tempering SMC brings two extra advantages: it makes it possible to obtain samples from $\pi_{\xi,\gamma}(\theta|\mathcal{D})$ for a whole range of values of $\gamma$, rather than a single value. And it provides an approximation of $Z_{\xi,\gamma}(\mathcal{D})$ for the same range of $\gamma$ values, through the quantity $Z_t$ defined in Algorithm 1.

### 3.3 Expectation-Propagation (Gaussian prior)

The SMC sampler outlined in the previous section works fairly well, and we will use it as gold standard in our simulations. However, as any other Monte Carlo method, it may be too slow for large datasets. We now turn our attention to EP [Expectation-Propagation Minka, 2001], a general framework to derive fast approximations to target distributions (and their normalising constants).

First note that the pseudo-posterior may be rewritten as:

$$\pi_{\xi,\gamma}(\theta|\mathcal{D}) = \frac{1}{Z_{\xi,\gamma}(\mathcal{D})}\pi_\xi(\theta) \times \prod_{i,j} f_{ij}(\theta), \quad f_{ij}(\theta) = \exp\left[-\gamma'\mathbb{1}\{\langle\theta, X_i - X_j\rangle < 0\}\right]$$

where $\gamma' = \gamma/n_+n_-$, and the product is over all $(i,j)$ such that $Y_i = 1$, $Y_j = -1$. EP generates an approximation of this target distribution based on the same factorisation:

$$q(\theta) \propto q_0(\theta) \prod_{i,j} q_{ij}(\theta), \quad q_{ij}(\theta) = \exp\{-\frac{1}{2}\theta^T Q_{ij}\theta + r_{ij}^T\theta\}.$$

We consider in the section the case where the prior is Gaussian, as in Section 2.3. Then one may set $q_0(\theta) = \pi_\xi(\theta)$. The approximating factors are un-normalised Gaussian densities (under a natural parametrisation), leading to an overall approximation that is also Gaussian, but other types of exponential family parametrisations may be considered; see next section and Seeger [2005]. EP updates iteratively each site $q_{ij}$ (that is, it updates the parameters $Q_{ij}$ and $r_{ij}$), conditional on all the sites, by matching the moments of $q$ with those of the hybrid distribution

$$h_{ij}(\theta) \propto q(\theta)\frac{f_{ij}(\theta)}{q_{ij}(\theta)} \propto q_0(\theta)f_{ij}(\theta) \prod_{(k,l)\neq(i,j)} f_{kl}(\theta)$$

where again the product is over all $(k, l)$ such that $Y_k = 1$, $Y_l = -1$, and $(k, l) \neq (i, j)$.

We refer to the supplement for a precise algorithmic description of our EP implementation. We highlight the following points. First, the site update is particularly simple in our case:

$$h_{ij}(\theta) \propto \exp\{\theta^T r_{ij}^h - \frac{1}{2}\theta^T Q_{ij}^h \theta\} \exp\left[-\gamma' \mathbb{1}\{\langle \theta, X_i - X_j \rangle < 0\}\right],$$

with $r_{ij}^h = \sum_{(k,l) \neq (i,j)} r_{kl}$, $Q_{ij}^h = \sum_{(k,l) \neq (i,j)} Q_{kl}$, which may be interpreted as: $\theta$ conditional on $T(\theta) = \langle \theta, X_i - X_j \rangle$ has a $d - 1$-dimensional Gaussian distribution, and the distribution of $T(\theta)$ is that of a one-dimensional Gaussian penalised by a step function. The two first moments of this particular hybrid may therefore be computed exactly, and in $\mathcal{O}(d^2)$ time, as explained in the supplement. The updates can be performed efficiently using the fact that the linear combination $(X_i - X_j)\theta$ is a one dimensional Gaussian. For our numerical experiment we used a parallel version of EP Van Gerven et al. [2010]. The complexity of our EP implementation is $\mathcal{O}(n_+ n_- d^2 + d^3)$.

Second, EP offers at no extra cost an approximation of the normalising constant $Z_{\xi,\gamma}(\mathcal{D})$ of the target $\pi_{\xi,\gamma}(\theta|\mathcal{D})$; in fact, one may even obtain derivatives of this approximated quantity with respect to hyper-parameters. See again the supplement for more details.

Third, in the EP framework, cross-validation may be interpreted as dropping all the factors $q_{ij}$ that depend on a given data-point $X_i$ in the global approximation $q$. This makes it possible to implement cross-validation at little extra cost [Opper and Winther, 2000].

### 3.4 Expectation-Propagation (spike and slab prior)

To adapt our EP algorithm to the spike and slab prior of Section 2.4, we introduce latent variables $Z_k = 0/1$ which "choose" for each component $\theta_k$ whether it comes from a slab, or from a spike, and we consider the joint target

$$\pi_{\xi,\gamma}(\theta, z|\mathcal{D}) \propto \left\{ \prod_{k=1}^{d} \mathcal{B}(z_k; p)\mathcal{N}(\theta_k; 0, v_{z_k}) \right\} \exp\left[ -\frac{\gamma}{n_+ n_-} \sum_{ij} \mathbb{1}\{\langle \theta, X_i - X_j \rangle > 0\} \right].$$

On top of the $n_+ n_-$ Gaussian sites defined in the previous section, we add a product of $d$ sites to approximate the prior. Following Hernandez-Lobato et al. [2013], we use

$$q_k(\theta_k, z_k) = \exp\left\{ z_k \log\left( \frac{p_k}{1 - p_k} \right) - \frac{1}{2}\theta_k^2 u_k + v_k \theta_k \right\}$$

that is a (un-normalised) product of an independent Bernoulli distribution for $z_k$, times a Gaussian distribution for $\theta_k$. Again that the site update is fairly straightforward, and may be implemented in $\mathcal{O}(d^2)$ time. See the supplement for more details. Another advantage of this formulation is that we obtain a Bernoulli approximation of the marginal pseudo-posterior $\pi_{\xi,\gamma}(z_i = 1|\mathcal{D})$ to use in feature selection. Interestingly taking $v_0$ to be exactly zero also yield stable results corresponding to the case where the spike is a Dirac mass.

## 4 Extension to non-linear scores

To extend our methodology to non-linear score functions, we consider the pseudo-posterior

$$\pi_{\xi,\gamma}(\mathrm{d}s|\mathcal{D}) \propto \pi_\xi(\mathrm{d}s) \exp\left\{ -\frac{\gamma}{n_+ n_-} \sum_{i \in \mathcal{D}_+, j \in \mathcal{D}_-} \mathbb{1}\{s(X_i) - s(X_j) > 0\} \right\}$$

where $\pi_\xi(\mathrm{d}s)$ is some prior probability measure with respect to an infinite-dimensional functional class. Let $s_i = s(X_i)$, $s_{1:n} = (s_1, \ldots, s_n) \in \mathbb{R}^n$, and assume that $\pi_\xi(\mathrm{d}s)$ is a GP (Gaussian process) associated to some kernel $k_\xi(x, x')$, then using a standard trick in the GP literature [Rasmussen and Williams, 2006], one may derive the marginal (posterior) density (with respect to

the $n$-dimensional Lebesgue measure) of $s_{1:n}$ as

$$\pi_{\xi,\gamma}(s_{1:n}|\mathcal{D}) \propto \mathcal{N}_d\left(s_{1:n}; 0, K_\xi\right) \exp\left\{-\frac{\gamma}{n_+ n_-} \sum_{i\in\mathcal{D}_+, j\in\mathcal{D}_-} \mathbb{1}\{s_i - s_j > 0\}\right\}$$

where $\mathcal{N}_d\left(s_{1:n}; 0, K_\xi\right)$ denotes the probability density of the $\mathcal{N}(0, K_\xi)$ distribution, and $K_\xi$ is the $n \times n$ matrix $\left(k_\xi(X_i, X_j)\right)_{i,j=1}^n$.

This marginal pseudo-posterior retains essentially the structure of the pseudo-posterior $\pi_{\xi,\gamma}(\theta|\mathcal{D})$ for linear scores, except that the "parameter" $s_{1:n}$ is now of dimension $n$. We can apply straightforwardly the SMC sampler of Section 3.2, and the EP algorithm of 3.3, to this new target distribution. In fact, for the EP implementation, the particular simple structure of a single site:

$$\exp\left[-\gamma' \mathbb{1}\{s_i - s_j > 0\}\right]$$

makes it possible to implement a site update in $\mathcal{O}(1)$ time, leading to an overall complexity $\mathcal{O}(n_+ n_- + n^3)$ for the EP algorithm.

Theoretical results for this approach could be obtained by applying lemmas from e.g. van der Vaart and van Zanten [2009], but we leave this for future study.

## 5   Numerical Illustration

Figure 1 compares the EP approximation with the output of our SMC sampler, on the well-known Pima Indians dataset and a Gaussian prior. Marginal first and second order moments essentially match; see the supplement for further details. The subsequent results are obtained with EP.

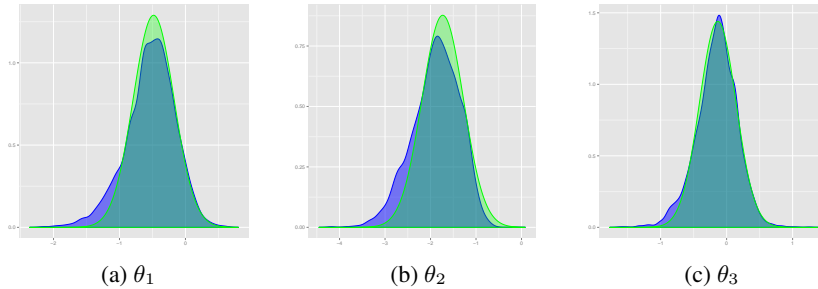

| (a) $\theta_1$ | (b) $\theta_2$ | (c) $\theta_3$ |

Figure 1: EP Approximation (green), compared to SMC (blue) of the marginal posterior of the first three coefficients, for Pima dataset (see the supplement for additional analysis).

We now compare our PAC-Bayesian approach (computed with EP) with Bayesian logistic regression (to deal with non-identifiable cases), and with the rankboost algorithm [Freund et al., 2003] on different datasets[1]; note that Cortes and Mohri [2003] showed that the function optimised by rankbook is AUC.

As mentioned in Section 3, we set the prior hyperparameters by maximizing the evidence, and we use cross-validation to choose $\gamma$. To ensure convergence of EP, when dealing with difficult sites, we use damping [Seeger, 2005]. The GP version of the algorithm is based on a squared exponential kernel. Table 1 summarises the results; balance refers to the size of the smaller class in the data (recall that the AUC criterion is particularly relevant for unbalanced classification tasks), EP-AUC (resp. GPEP-AUC) refers to the EP approximation of the pseudo-posterior based on our Gaussian prior (resp. Gaussian process prior). See also Figure 2 for ROC curve comparisons, and Table 1 in the supplement for a CPU time comparison.

Note how the GP approach performs better for the colon data, where the number of covariates (2000) is very large, but the number of observations is only 40. It seems also that EP gives a better approximation in this case because of the lower dimensionality of the pseudo-posterior (Figure 2b).

| Dataset | Covariates | Balance | EP-AUC | GPEP-AUC | Logit | Rankboost |
|---------|-----------|---------|--------|----------|-------|-----------|
| Pima | 7 | 34% | 0.8617 | 0.8557 | **0.8646** | 0.8224 |
| Credit | 60 | 28% | **0.7952** | 0.7922 | 0.7561 | 0.788 |
| DNA | 180 | 22% | **0.9814** | 0.9812 | 0.9696 | **0.9814** |
| SPECTF | 22 | 50% | 0.8684 | 0.8545 | **0.8715** | 0.8684 |
| Colon | 2000 | 40% | 0.7034 | **0.75** | 0.73 | 0.5935 |
| Glass | 10 | 1% | **0.9843** | 0.9629 | 0.9029 | 0.9436 |

Table 1: Comparison of AUC.
The Glass dataset has originally more than two classes. We compare the "silicon" class against all others.

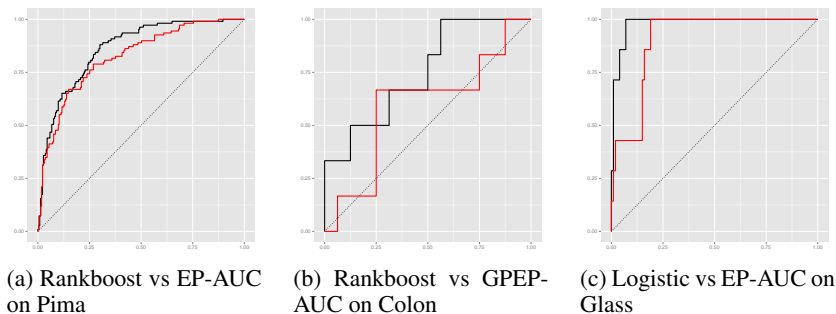

(a) Rankboost vs EP-AUC on Pima

(b) Rankboost vs GPEP-AUC on Colon

(c) Logistic vs EP-AUC on Glass

Figure 2: Some ROC curves associated to the example described in a more systematic manner in table 1. In black is always the PAC version.

Finally, we also investigate feature selection for the DNA dataset (180 covariates) using a spike and slab prior. The regularization plot (3a) shows how certain coefficients shrink to zero as the spike's variance $v_0$ goes to zero, allowing for some sparsity. The aim of a positive variance in the spike is to absorb negligible effects into it [Ročková and George, 2013]. We observe this effect on figure 3a where one of the covariates becomes positive when $v_0$ decreases.

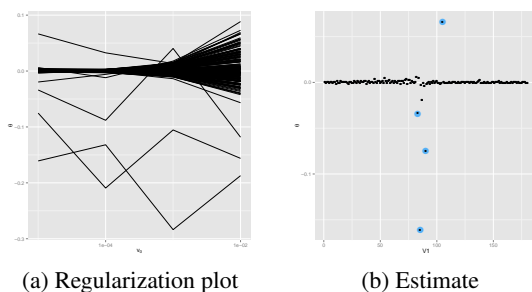

(a) Regularization plot

(b) Estimate

Figure 3: Regularization plot for $v_0 \in \left[10^{-6}, 0.1\right]$ and estimation for $v_0 = 10^{-6}$ for DNA dataset; blue circles denote posterior probabilities $\geq 0.5$.

## 6  Conclusion

The combination of the PAC-Bayesian theory and Expectation-Propagation leads to fast and efficient AUC classification algorithms, as observed on a variety of datasets, some of them very unbalanced. Future work may include extending our approach to more general ranking problems (e.g. multiclass), establishing non-asymptotic bounds in the nonparametric case, and reducing the CPU time by considering only a subset of all the pairs of datapoints.

## Footnotes

[1]All available at http://archive.ics.uci.edu/ml/

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
