[Supplementary Material]

# Supplement to: PAC-Bayesian AUC classification and scoring

**James Ridgway**[*]
CREST and CEREMADE University Dauphine
Timbre J120
3, Avenue Pierre Larousse 92245 MALAKOFF CEDEX
FRANCE
james.ridgway@ensae.fr

**Pierre Alquier**
University College Dublin
Address
pierre.alquier@ucd.ie

**Nicolas Chopin**
CREST(ENSAE) and HEC Paris
Timbre J120
3, Avenue Pierre Larousse 92245 MALAKOFF CEDEX
FRANCE
nicolas.chopin@ensae.fr

**Feng Liang**
University of Illinois at Urbana-Champaign
116A Illini Hall 725 S. Wright St. Champaign, IL 61820 USA

## Abstract

This supplement follows the same plan as the paper.

## 2 PAC-Bayes bounds for linear scores

### 2.2 Assumptions and general results

A simple sufficient condition for Dens(c) to hold is that $(X_1 - X_2)/\|X_1 - X_2\|$ admits a probability density with respect to the spherical measure of dimension $d-1$ which is bounded above by $B$. Then

$$\mathbb{P}(\langle X_1 - X_2, \theta \rangle \geq 0, \langle X_1 - X_2, \theta' \rangle \leq 0) \leq B \frac{\arccos(\langle \theta, \theta' \rangle)}{2\pi}$$

$$\leq \frac{B}{2\pi} \sqrt{5 - 5 \langle \theta, \theta' \rangle}$$

$$= \frac{B}{2\pi} \sqrt{\frac{5}{2}} \|\theta - \theta'\|.$$

**Proof of lemma 2.1**

In order to prove lemma 2.1 we need the following Bernstein inequality.

**Proposition 2.1 (Bernstein's inequality for U-statistics)** *For any $\gamma > 0$, for any $\theta \in \mathbb{R}^d$,*

$$\mathbb{E} \exp[\gamma | R_n(\theta) - \overline{R}_n - R(\theta) + \overline{R} |] \leq 2 \exp \left[ \frac{\frac{\gamma^2}{n-1} \mathbb{E}((q_{1,2}^\theta)^2)}{\left( 1 - \frac{4\gamma}{n-1} \right)} \right].$$

[*]http://www.crest.fr/pagesperso.php?user=3328

*Proof of Proposition 2.1.* Fix $\theta$. Remember that

$$q_{i,j}^{\theta} = \mathbf{1}\{\langle \theta, X_i - X_j \rangle (Y_i - Y_j) < 0\} - \mathbf{1}\{[\sigma(X_i) - \sigma(X_j)](Y_i - Y_j) < 0\} - R(\theta) + \overline{R}$$

so that

$$U_n := R_n(\theta) - \overline{R}_n - R(\theta) + \overline{R} = \frac{1}{n(n-1)} \sum_{i \neq j} q_{i,j}^{\theta}.$$

First, note that

$$\mathbb{E} \exp[\gamma |U_n|] \leq \mathbb{E} \exp[\gamma U_n] + \mathbb{E} \exp[\gamma(-U_n)].$$

We will only upper bound the first term in the r.h.s., as the upper bound for the second term may be obtained exactly in the same way (just replace $q_{i,j}^{\theta}$ by $-q_{i,j}^{\theta}$). Now, use Hoeffding's decomposition Hoeffding [1948]: this is the technique used by Hoeffding to prove inequalities on U-statistics. Hoeffding proved that

$$U_n = \frac{1}{n!} \sum_{\pi} \frac{1}{\lfloor \frac{n}{2} \rfloor} \sum_{i=1}^{\lfloor \frac{n}{2} \rfloor} q_{\pi(i),\pi(i+\lfloor \frac{n}{2} \rfloor)}^{\theta}$$

where the sum is taken over all the permutations $\pi$ of $\{1, \ldots, n\}$. Jensen's inequality leads to

$$\mathbb{E} \exp[\gamma U_n] = \mathbb{E} \exp \left[ \gamma \frac{1}{n!} \sum_{\pi} \frac{1}{\lfloor \frac{n}{2} \rfloor} \sum_{i=1}^{\lfloor \frac{n}{2} \rfloor} q_{\pi(i),\pi(i+\lfloor \frac{n}{2} \rfloor)}^{\theta} \right]$$

$$\leq \frac{1}{n!} \sum_{\pi} \mathbb{E} \exp \left[ \frac{\gamma}{\lfloor \frac{n}{2} \rfloor} \sum_{i=1}^{\lfloor \frac{n}{2} \rfloor} q_{\pi(i),\pi(i+\lfloor \frac{n}{2} \rfloor)}^{\theta} \right].$$

We now use, for each of the terms in the sum, Massart's version of Bernstein's inequality Massart [2007] (ineq. (2.21) in Chapter 2, the assumption is checked by $q_{\pi(i),\pi(i+\lfloor \frac{n}{2} \rfloor)}^{\theta} \in [-2, 2]$ so $\mathbb{E}((q_{\pi(i),\pi(i+\lfloor \frac{n}{2} \rfloor)}^{\theta})^k) \leq \mathbb{E}((q_{\pi(i),\pi(i+\lfloor \frac{n}{2} \rfloor)}^{\theta})^2) 2^{k-2}$). We obtain:

$$\mathbb{E} \exp \left[ \frac{\gamma}{\lfloor \frac{n}{2} \rfloor} \sum_{i=1}^{\lfloor \frac{n}{2} \rfloor} q_{\pi(i),\pi(i+\lfloor \frac{n}{2} \rfloor)}^{\theta} \right] \leq \exp \left[ \frac{\mathbb{E}((q_{\pi(1),\pi(1+\lfloor \frac{n}{2} \rfloor)}^{\theta})^2) \frac{\gamma^2}{\lfloor \frac{n}{2} \rfloor}}{2 \left( 1 - 2\frac{\gamma}{\lfloor \frac{n}{2} \rfloor} \right)} \right].$$

First, note that we have the inequality $\lfloor \frac{n}{2} \rfloor \geq (n-1)/2$. Then, remark that as the pairs $(X_i, Y_i)$ are iid, we have $\mathbb{E}((q_{\pi(1),\pi(1+\lfloor \frac{n}{2} \rfloor)}^{\theta})^2) = \mathbb{E}((q_{1,2}^{\theta})^2)$ so we have a simpler inequality

$$\mathbb{E} \exp \left[ \frac{\gamma}{\lfloor \frac{n}{2} \rfloor} \sum_{i=1}^{\lfloor \frac{n}{2} \rfloor} q_{\pi(i),\pi(i+\lfloor \frac{n}{2} \rfloor)}^{\theta} \right] \leq \exp \left[ \frac{\mathbb{E}((q_{1,2}^{\theta})^2) \frac{\gamma^2}{n-1}}{\left( 1 - \frac{4\gamma}{n-1} \right)} \right].$$

This ends the proof of the proposition. $\square$
The following proposition is also of use in the proof of lemma 2.1.

**Proposition 2.2** *For any measure $\rho \in \mathcal{M}_+^1(\Theta)$ and any measurable function $h : \theta \to \mathbb{R}$ such that $\int \exp(h(\theta))\pi(\mathrm{d}\theta) < \infty$, we have*

$$\log \left( \int \exp(h(\theta))\pi(\theta) \right) = \sup_{\rho \in \mathcal{M}_+^1} \left( \int h(\theta)\rho(\mathrm{d}\theta) - \mathcal{K}(\rho, \pi) \right).$$

*In addition if $h$ is bounded by above on the support of $\pi$ the supremum is reached for the Gibbs distribution,*

$$\rho(\mathrm{d}\theta) \propto \exp(h(\theta)) \pi(\mathrm{d}\theta).$$

**Proof:** e.g. Catoni [2007]. $\square$

**Proof of Lemma 2.1** From the proof of Proposition 2.1, and using the short-hand $q_\theta$ for $q_{1,2}^\theta$, we deduce

$$\mathbb{E}\left[\exp\{\rho\left(\gamma(R_n(\theta) - \bar{R}_n - R(\theta) + \bar{R})\} + \eta(\theta))\right] \le \exp\left(\frac{\gamma^2}{n-1}\frac{\rho\left(\mathbb{E}q_\theta^2\right)}{(1 - 4\frac{\gamma}{n-1})} + \rho\left(\eta(\theta)\right)\right). \tag{1}$$

Using proposition 2.2, and the fact that $e^x \ge \mathbb{1}\{x \ge 0\}$ we have that

$$\mathbb{P}\{\sup_{\rho \in \mathcal{M}_+^1(\Theta)} \rho\left(\gamma(R_n(\theta) - \bar{R}_n - R(\theta) + \bar{R}) - \eta(\theta)\right) - \mathcal{K}(\rho, \pi) \ge 0\}$$

$$\le \mathbb{E}\left(\pi\{\exp\{\rho\left(\gamma(R_n(\theta) - \bar{R}_n - R(\theta) + \bar{R}) - \eta(\theta)\})\right)\}\right)$$

$$= \pi\left(\mathbb{E}\{\exp\{\rho\left(\gamma(R_n(\theta) - \bar{R}_n - R(\theta) + \bar{R}) - \eta(\theta)\})\right)\}\right) \quad \text{, by Fubini}$$

$$\le \pi\left\{\exp\left(\frac{\gamma^2 \rho(\mathbb{E}q_\theta^2)}{(n-1)(1 - \frac{4\gamma}{n-1})} - \rho(\eta(\theta))\right)\right\} \quad \text{, using (1).}$$

In the following we take $\eta(\theta) = \log\frac{1}{\epsilon} + \frac{\gamma^2}{n-1}\frac{\rho(\mathbb{E}q_\theta^2)}{(1 - 4\frac{\gamma}{n-1})}$ leading to the following result with probability at least $1 - \epsilon$, $\forall \rho \in \mathcal{M}_+^1(\Theta)$:

$$\rho(R_n(\theta)) - \bar{R}_n \le \rho(R(\theta)) - \bar{R} + \frac{\mathcal{K}(\rho, \pi) + \log\frac{1}{\epsilon}}{\gamma} + \frac{\gamma}{n-1}\frac{\rho(\mathbb{E}q_\theta^2)}{(1 - 4\frac{\gamma}{n-1})}. \tag{2}$$

Under **MA**$(1, C)$ we can write:

$$\rho(R_n(\theta)) - \bar{R}_n \le \left(1 + \frac{\gamma C}{n-1}\frac{1}{(1 - \frac{4}{n-1})}\right)\left(\rho(R(\theta)) - \bar{R}\right) + \frac{\mathcal{K}(\rho, \pi) + \log\frac{1}{\epsilon}}{\gamma}.$$

Using Bernstein's inequality in the symmetric case, with probability $1 - \epsilon$ we can assert that:

$$\left(1 - \frac{\gamma C}{n-1}\frac{1}{(1 - \gamma\frac{4}{n-1})}\right)\left(\rho(R(\theta)) - \bar{R}\right) \le \rho(R_n(\theta)) - \bar{R}_n + \frac{\mathcal{K}(\rho, \pi) + \log\frac{1}{\epsilon}}{\gamma}.$$

The latter is true in particular for $\rho = \pi(\theta|\mathcal{S})$, the Gibbs posterior:

$$\left(1 - \frac{\gamma C}{n-1}\frac{1}{(1 - \gamma\frac{4}{n-1})}\right)\left(\int_\Theta R(\theta)\pi_\gamma(d\theta|\mathcal{D}) - \bar{R}\right) \le \inf_{\rho \in \mathcal{M}_+^1}\left\{\rho(R_n(\theta)) - \bar{R}_n + \frac{\mathcal{K}(\rho, \pi) + \log\frac{1}{\epsilon}}{\gamma}\right\}.$$

Making use of equation (2) and the fact that $\gamma \le (n-1)/8C$ we have with probability $1 - 2\epsilon$:

$$\left(\int_\Theta R_n(\theta)\pi_\gamma(d\theta|\mathcal{D}) - \bar{R}_n\right) \le 2\inf_{\rho \in \mathcal{M}_+^1}\left(\rho(R(\theta)) - \bar{R} + 2\frac{\mathcal{K}(\rho, \pi) + \log\frac{1}{\epsilon}}{\gamma}\right). \qquad \square$$

Lemma 2.1 gives some approximately correct finite sample bound under hypothesis **MA**$(1, C)$. It is easy to extend those results to the more general case of **MA**$(\infty, C)$. Note in particular that this assumption is always satisfied for $C = 4$.

**Proof of Lemma 2.2** First consider in our case that, the margin assumption is always true for $C = 4$, $\mathbb{E}(q_\theta^2) \le 4$, the rest of the proof is similar to that of lemma 2.1. From equation (2) with the above hypothesis:

$$\rho(R_n(\theta)) - \bar{R}_n \le \rho(R(\theta)) - \bar{R} + \frac{\mathcal{K}(\rho, \pi) + \log\frac{1}{\epsilon}}{\gamma} + \frac{\gamma}{n-1}\frac{4}{(1 - \frac{4}{n-1})}$$

From the Bernstein inequality with in the symmetric case we get with probability $1 - \epsilon$:

$$\rho(R(\theta)) - \bar{R} \le \rho(R_n(\theta)) - \bar{R}_n + \frac{\mathcal{K}(\rho, \pi) + \log\frac{1}{\epsilon}}{\gamma} + \frac{\gamma}{n-1}\frac{4}{(1 - \frac{4\gamma}{n-1})}$$

We get, after noting that the Gibbs posterior can be written as an infimum (Legendre transform), with probability $1 - 2\epsilon$:

$$\int (R(\theta)\pi_\gamma(d\theta|\mathcal{D}) - \bar{R} \leq \inf_{\rho \in \mathcal{M}_+^1(\Theta)} \rho(R(\theta)) - \bar{R} + 2\frac{\mathcal{K}(\rho,\pi) + \log\frac{1}{\epsilon}}{\gamma} + \frac{16\gamma}{n-1}$$

(we also used $\gamma \leq (n-1)/8$).

$\square$

The two above lemma depend on some class complexity $\mathcal{K}(\rho,\pi)$. The latter can be specialized to different choice of prior measure $\pi$. In the following we propose two specifications to a Gaussian prior and a spike and slab prior.

## 2.3 Independent Gaussian Prior

**Theorem 2.3** *Assume* $\mathbf{MA}(1,C)$, $C \geq 1$, $\mathbf{Dens}(c)$, $c > 0$, *and take* $\vartheta = \frac{2}{d}(1 + \frac{1}{n^2 d})$, $\gamma = (n-1)/8C$, *then there exists a constant* $\alpha = \alpha(c,C,d)$ *such that for any* $\epsilon > 0$, *with probability* $1 - \epsilon$,

$$\int R(\theta)\pi_\gamma(\theta|\mathcal{D})d\theta - \bar{R} \leq 2\inf_{\theta_0}\left(R(\theta_0) - \bar{R} + \alpha\frac{d\log(n) + \log\frac{4}{\epsilon}}{n-1}\right).$$

**Proof:** For any $\theta_0 \in \mathbb{R}^p$ with $\|\theta_0\| = 1$ and $\delta > 0$ we put

$$\rho_{\theta_0,\delta}(d\theta) \propto \mathbf{1}_{\|\theta-\theta_0\|\leq\delta}\pi(d\theta).$$

Then we have, from Lemma 2.1, with probability at least $1 - \varepsilon$,

$$\int R(\theta)\pi_\gamma(d\theta|\mathcal{D}) - \overline{R} \leq 2\inf_{\theta_0,\delta}\left\{\int R(\theta)\rho_{\theta_0,\delta}(d\theta) - \overline{R} + 16C\frac{\mathcal{K}(\rho_{\theta_0,\delta},\pi) + \log\left(\frac{4}{\varepsilon}\right)}{(n-1)}\right\}$$

First, note that

$$
\begin{aligned}
R(\theta) &= \mathbb{E}\left(\mathbb{1}\{\langle\theta, X - X'\rangle(Y - Y') < 0\}\right)\\
&= \mathbb{E}\left(\mathbb{1}\{\langle\theta_0, X - X'\rangle(Y - Y') < 0\}\right)\\
&\quad + \mathbb{E}\left(\mathbb{1}\{\langle\theta, X - X'\rangle(Y - Y') < 0\} - \mathbb{1}\{\langle\theta_0, X - X'\rangle(Y - Y') < 0\}\right)\\
&\leq R(\theta_0) + \mathbb{P}(\text{sign}\,\langle\theta, X - X'\rangle(Y - Y') \neq \text{sign}\,\langle\theta_0, X - X'\rangle(Y - Y'))\\
&= R(\theta_0) + \mathbb{P}(\text{sign}\,\langle\theta, X - X'\rangle \neq \text{sign}\,\langle\theta_0, X - X'\rangle)\\
&\leq R(\theta_0) + c\left\|\frac{\theta}{\|\theta\|} - \theta_0\right\|\\
&\leq R(\theta_0) + 2c\|\theta - \theta_0\|.
\end{aligned}
$$

As a consequence $\int R(\theta)\rho_{\theta_0,\delta}(d\theta) \leq R(\theta_0) + 2c\delta$.

The next step is to calculate $\mathcal{K}(\rho_{\theta_0,\delta},\pi)$. We have

$$\mathcal{K}(\rho_{\theta_0,\delta},\pi) = \log\frac{1}{\pi\left(\{\theta : \|\theta - \theta_0\| \leq \delta\}\right)}.$$

Assuming that $\theta_{0,1} > 0$ (the proof is exactly symmetric in the other case)

$$-\mathcal{K}(\rho_{\theta_0,\delta}, \pi) = \log \pi \left( \{\theta : \sum_{i=1}^{d}(\theta_i - \theta_{0,i})^2 \leq \delta^2\} \right)$$

$$\geq d \log \pi \left( \{\theta : (\theta_1 - \theta_{0,1})^2 \leq \frac{\delta^2}{d}\} \right)$$

$$\geq d \log \int_{\frac{\theta_{0,1}}{\sqrt{\vartheta}} - \frac{\delta}{\sqrt{\vartheta d}}}^{\frac{\theta_{0,1}}{\sqrt{\vartheta}} + \frac{\delta}{\sqrt{\vartheta d}}} \varphi_{(0,1)}(x) \mathrm{d}x$$

$$\geq d \log \left( \frac{\delta}{2\sqrt{\vartheta d}} \varphi \left( \frac{\theta_{0,1}}{\sqrt{\vartheta}} + \frac{\delta}{\sqrt{\vartheta d}} \right) \right)$$

$$\geq d \log \left( \frac{\delta}{2\sqrt{\vartheta d}} \varphi \left( \frac{1}{\sqrt{\vartheta}} + \frac{\delta}{\sqrt{\vartheta d}} \right) \right)$$

$$= d \log \left( \frac{\delta}{2\sqrt{2\pi \vartheta d}} \exp \left[ -\frac{1}{2} \left( \frac{1}{\sqrt{\vartheta}} + \frac{\delta}{\sqrt{\vartheta d}} \right)^2 \right] \right)$$

$$\geq d \log \left\{ \frac{\delta}{2\sqrt{2\pi \vartheta d}} \exp \left( -\frac{1}{\vartheta} - \frac{\delta^2}{\vartheta d} \right) \right\}$$

$$\mathcal{K}(\rho_{\theta_0,\delta}, \pi) \leq -d \log\{\delta\} + \frac{d}{2}\log\{8\pi\vartheta d\} + \frac{1}{\vartheta} + \frac{\delta^2}{\vartheta d}$$

And we can plug the equation above in the result of lemma 2.1 with $\delta = \frac{1}{n}$

$$\int R(\theta)\pi_\gamma(\theta|\mathcal{D}) - \bar{R} \leq 2 \inf_{\theta_0} \left( R(\theta_0) - \bar{R} + 2c\frac{1}{n} + \frac{2}{\gamma} \left( d\log\{n\} + \frac{d}{2}\log\{8\pi\vartheta d\} + \frac{1}{\vartheta} + \frac{\frac{1}{n^2}}{\vartheta d} + \log\frac{4}{\epsilon} \right) \right)$$

Any $\gamma = O(n)$ will lead to a convergence result. Taking $\gamma = (n-1)/8C$ and optimizing in $\vartheta$ we obtain a variance of $\vartheta = \frac{2(1+\frac{1}{n^2 d})}{d}$. $\qquad\square$

As was done for the previous lemmas we can lift the **MA**$(\infty, C)$ and use the lemma 2.2 instead, this gives rise to the following theorem.

**Theorem 2.4** *Assume* **MA**$(\infty, C)$, $C \geq 1$, **Dens**$(c)$ $c > 0$, *and take* $\vartheta = \frac{2}{d}(1 + \frac{1}{n^2 d})$, $\gamma = C\sqrt{dn \log(n)}$, *there exists a constant* $\alpha = \alpha(c, C, d)$ *such that for any* $\epsilon > 0$, *with probability* $1 - \epsilon$,

$$\int R(\theta)\pi_\gamma(\theta|\mathcal{D})\mathrm{d}\theta - \bar{R} \leq \inf_{\theta_0} \left( R(\theta_0) - \bar{R} + \alpha \frac{\sqrt{d\log(n)} + \log\frac{2}{\epsilon}}{\sqrt{n}} \right).$$

**Proof:** Use Lemma 2.2 and the same steps as in the proof of Theorem 2.3, optimize w.r.t. $\gamma$ and $\vartheta$ to get the result. $\qquad\square$

We show the same kind of result in the following but for spike and slab priors.

## 2.4 Spike and slab prior for feature selection

**Theorem 2.5** *Assume* **MA**$(1, C)$ *holds with* $C \geq 1$, **Dens**$(c)$ *holds with* $c > 0$, *and take* $p = 1 - \exp(-1/d)$ *and* $v_0 \leq 1/(2nd\log(d))$, *and* $\gamma = (n-1)/(8C)$. *Then there is a constant* $\alpha = \alpha(C, v_1, c)$ *such that for any* $\varepsilon > 0$, *with probability at least* $1 - \varepsilon$ *on the drawing of the data* $\mathcal{D}$,

$$\int R(\theta)\pi_\gamma(\mathrm{d}\theta|\mathcal{D}) - \overline{R} \leq 2 \inf_{\theta_0} \left\{ R(\theta_0) - \overline{R} + \alpha \frac{\|\theta_0\|_0 \log(nd) + \log\left(\frac{4}{\varepsilon}\right)}{2(n-1)} \right\}.$$

**Proof:** As for the proof of theorem 2.3 we start by defining, for any $\theta_0 \in \mathbb{R}^p$ with $\|\theta_0\| = 1$ and $\delta > 0$,

$$\rho_{\theta_0,\delta}(\mathrm{d}\theta) \propto \mathbf{1}_{\|\theta - \theta_0\| \leq \delta} \pi(\mathrm{d}\theta)$$

so that in the end, by a similar argument as previously it remains only to upper bound the following quantity,

$$\mathcal{K}(\rho_{\theta_0,\delta}, \pi) = \log \frac{1}{\pi(\{\theta : \|\theta - \theta_0\| \leq \delta\})}.$$

Let $\pi_0$ denote the probability distribution such that the $\theta_i$ are iid $\mathcal{N}(0, v_0)$. So:

$$
\begin{aligned}
-\mathcal{K}(\rho_{\theta_0,\delta}, \pi) &= \log \pi \left( \left\{ \theta : \sum_{i=1}^d (\theta_i - \theta_{0,i})^2 \leq \delta^2 \right\} \right) \\
&\geq \log \pi \left( \left\{ \theta : \forall i, (\theta_i - \theta_{0,i})^2 \leq \frac{\delta^2}{d} \right\} \right) \\
&= \sum_{i:\theta_{0,i} \neq 0} \log \pi \left( \left\{ (\theta_i - \theta_{0,i})^2 \leq \frac{\delta^2}{d} \right\} \right) \\
&\quad + \log \pi \left( \left\{ \forall i \text{ with } \theta_{0,i} = 0, \theta_i^2 < \frac{\delta^2}{d} \right\} \right) \\
&\geq \sum_{i:\theta_{0,i} \neq 0} \log \pi \left( \left\{ (\theta_i - \theta_{0,i})^2 \leq \frac{\delta^2}{d} \right\} \right) \\
&\quad + \log \pi_0 \left( \left\{ \forall i \text{ with } \theta_{0,i} = 0, \theta_i^2 < \frac{\delta^2}{d} \right\} \right) + d \log(1 - p) \\
&= \sum_{i:\theta_{0,i} \neq 0} \log \pi \left( \left\{ (\theta_i - \theta_{0,i})^2 \leq \frac{\delta^2}{d} \right\} \right) \\
&\quad + \log \left[ 1 - \pi_0 \left( \left\{ \exists i, \theta_{0,i} = 0, \theta_i^2 > \frac{\delta^2}{d} \right\} \right) \right] + d \log(1 - p) \\
&\geq \sum_{i:\theta_{0,i} \neq 0} \log \pi \left( \left\{ (\theta_i - \theta_{0,i})^2 \leq \frac{\delta^2}{d} \right\} \right) \\
&\quad + \log \left[ 1 - \sum_{i:\theta_i = 0} \pi_0 \left( \left\{ \theta_i^2 > \frac{\delta^2}{d} \right\} \right) \right] + d \log(1 - p).
\end{aligned}
$$

Assume first that $i$ is such that $\theta_{0,i} = 0$. Then:

$$
\begin{aligned}
\pi_0 \left( \left\{ \theta_i^2 > \frac{\delta^2}{d} \right\} \right) &= \pi_0 \left( \left\{ \left| \frac{\theta_i}{\sqrt{v_0}} \right| > \frac{\delta}{\sqrt{v_0 d}} \right\} \right) \\
&\leq \exp \left( -\frac{\delta^2}{2 v_0 d} \right),
\end{aligned}
$$

and so

$$\sum_{i:\theta_{0,i}=0} \pi_0 \left( \left\{ \theta_i^2 > \frac{\delta^2}{d} \right\} \right) \leq d \exp \left( -\frac{\delta^2}{2 v_0 d} \right) \leq \frac{1}{2}$$

as soon as $v_0 \leq \delta^2/(2d\log(d))$. Then, assume that $i$ is such that $\theta_{0,i} \neq 0$. Now assume that $\theta_{0,i} > 0$ (the proof is exactly symmetric if $\theta_{0,i} < 0$):

$$\pi\left(\left\{\theta : (\theta_i - \theta_{0,i})^2 \leq \frac{\delta^2}{d}\right\}\right) \geq p \int_{\frac{\theta_{0,i}}{\sqrt{v_1}} - \frac{\delta}{\sqrt{v_1 d}}}^{\frac{\theta_{0,i}}{\sqrt{v_1}} + \frac{\delta}{\sqrt{v_1 d}}} \varphi_{(0,1)}(x)\mathrm{d}x$$

$$\geq \frac{p\delta}{2\sqrt{v_1 d}}\varphi\left(\frac{\theta_{0,i}}{\sqrt{v_1}} + \frac{\delta}{\sqrt{v_1 d}}\right)$$

$$\geq \frac{p\delta}{2\sqrt{v_1 d}}\varphi\left(\frac{1}{\sqrt{v_1}} + \frac{\delta}{\sqrt{v_1 d}}\right)$$

$$= \frac{p\delta}{2\sqrt{2\pi v_1 d}}\exp\left[-\frac{1}{2}\left(\frac{1}{\sqrt{v_1}} + \frac{\delta}{\sqrt{v_1 d}}\right)^2\right]$$

$$\geq \frac{p\delta}{2\sqrt{2\pi v_1 d}}\exp\left[-\frac{1}{v_1} - \frac{\delta^2}{v_1 d}\right].$$

Putting everything together:

$$\mathcal{K}(\rho_{\theta_0,\delta}, \pi) \leq -\|\theta_0\|_0 \log\left(\frac{p\delta}{2\sqrt{2\pi v_1 d}}\exp\left[-\frac{1}{v_1} - \frac{\delta^2}{v_1 d}\right]\right) + \log(2) + d\log\frac{1}{1-p}$$

$$= \|\theta_0\|_0\left[\log\left(\frac{2\sqrt{2\pi v_1 d}}{p\delta}\right) + \frac{1}{v_1} + \frac{\delta^2}{v_1 d}\right] + \log(2) + d\log\frac{1}{1-p}.$$

So, we have:

$$\int R(\theta)\pi_\gamma(\mathrm{d}\theta|\mathcal{D}) - \overline{R} \leq 2\inf_{\theta_0,\delta}\Bigg\{R(\theta_0) - \overline{R} + 2c\delta$$

$$+ 16C\frac{\|\theta_0\|_0\left[\log\left(\frac{2\sqrt{2\pi v_1 d}}{p\delta}\right) + \frac{1}{v_1} + \frac{\delta^2}{v_1 d}\right] + \log(2) + d\log\frac{1}{1-p} + \log\left(\frac{4}{\varepsilon}\right)}{(n-1)}\Bigg\}$$

□

# 3 Practical implementation of the PAC-Bayesian approach

## 3.2 Sequential Monte Carlo

The resampling scheme we use in our SMC sampler is systematic resampling, see Algorithm 1.

---
**Algorithm 1** Systematic resampling
---

**Input:** Normalised weights $W_t^j := w_t(\theta_{t-1}^j)/\sum_{i=1}^N w_t(\theta_{t-1}^i)$.

**Output:** indices $A^i \in \{1, \ldots, N\}$, for $i = 1, \ldots, N$.

**a.** Sample $U \sim \mathcal{U}([0,1])$.

**b.** Compute cumulative weights as $C^n = \sum_{m=1}^n NW^m$.

**c.** Set $s \leftarrow U, m \leftarrow 1$.

**d. For** $n = 1 : N$

    **While** $C^m < s$ **do** $m \leftarrow m + 1$.

    $A^n \leftarrow m$, and $s \leftarrow s + 1$.

**End For**

---

To move the particles while leaving invariant the current target $\pi_{\xi,\gamma}(\theta|\mathcal{D})$, we use the standard random walk Metropolis strategy, but scaled to the current set of particles, as outlined by Algorithm 2.

---
**Algorithm 2** Gaussian random walk Metropolis step
---

**Input:** $\theta$, $S$ ($d \times d$ positive matrix)

**Output:** $\theta_{\text{next}}$

**a.** Sample $\theta_{\text{prop}} \sim \mathcal{N}(\theta, S)$.

**b.** Sample $U \sim \mathcal{U}([0,1])$.

**c.** If $\log(U) \leq \log \pi_{\xi,\gamma}(\theta_{\text{prop}}|\mathcal{D})/\pi_{\xi,\gamma}(\theta|\mathcal{D})$, set $\theta_{\text{next}} \leftarrow \theta_{\text{prop}}$, otherwise set $\theta_{\text{next}} \leftarrow \theta$.

---

### 3.3 Expectation-Propagation (Gaussian prior)

EP aims at approximating posterior distributions of the form,

$$\pi(\theta|\mathcal{D}) = \frac{1}{Z_\pi} P_0(\theta) \prod_{i=1}^n t_i(\theta)$$

by approximating each site $t_i(\theta)$ by a distribution from an exponential family $q_i(\theta)$. The algorithm cycles through each site, computes the cavity distribution $Q^{\backslash i}(\theta) \propto Q(\theta)q_i^{-1}(\theta)$ and minimizes the Kullback-Leibler divergence between $Q^{\backslash i}(\theta)t_i(\theta)$ and the global approximation $Q(\theta)$. This is efficiently done by using properties of the exponential family (e.g. Bishop [2006]).

In the Gaussian case the EP approximation can be written as a product of some prior and a product of sites:

$$Q(\theta) \propto \mathcal{N}(\theta; 0, \Sigma) \prod_{i,j} q_{ij}(\theta),$$

for which the sites are unnormalized Gaussians for the natural parametrization $q_{ij}(\theta) \propto \exp\left(-\frac{1}{2}\theta^T Q_{ij}\theta + \theta r_{ij}\right)$. We can equivalently use the one dimensional representation $q_{ij}(s_{ij}) \propto \exp\left(-\frac{1}{2}s_{ij}^2 K_{ij} + s_{ij}h_{ij}\right)$, going from one to the other is easily done by multiplying $\theta$ by $(e_i - e_j)X$ where $\forall i \in \{1, \cdots, n\}$, $e_i$ is a vector of zeroes with one on the i-th line. Hence we keep in memory only $(K_{ij})_{ij}$ and $(h_{ij})_{ij}$.

While computing the cavity moment we must compute $(Q - (X_i - X_j)(X_i - X_j)K_{ij})$ and its inverse. The latter can be computed efficiently using Woodbury formula. Equivalently one could use similar tricks where only the Cholesky factorisation is saved and updated as in Seeger [2005]. By precomputing some matrix multiplication the later cavity moment computation can be done in complexity $\mathcal{O}(p^2)$.

To update the sites we compute normalizing constant $Z_{ij} = \int \mathcal{N}(s; m^{\backslash ij}, \sigma^{\backslash ij})t_{ij}(s)\mathrm{d}s$ and use properties of exponential families.

---
**Algorithm 3** parallel EP for Gaussian Prior
---

**Input:** $\vartheta, \gamma$

**Output:** $m$ and $V$

**Init:** $V \leftarrow \Sigma, m \leftarrow 0$

**Untill** Convergence **Do**

**For** all sites $(i,j)$**Do** in parallel

      a. Compute the cavity moments $m^{\backslash ij}, V^{\backslash ij}$

      b. Compute the 1st and 2nd order moments of $q^{\backslash ij}(s_{ij})t_{ij}(s_ij)$

      c. Update $K_{ij}$ and $h_{ij}$

**End For**

Update $V = (\Sigma^{-1} + \sum_{ij}(X_i - X_j)^T(X_i - X_j)K_{ij})^{-1}$, $m = V(\sum_{ij}(X_i - X_j)h_{ij})$

**End While**

---

**Normalising Constant**  The normalizing constant of the posterior can be computed using EP. We have that for each sites $t_{ij}(\theta) = C_{ij}q_{ij}(\theta)$ we replace those sites in integral we wish to approximate,

$$\int \mathcal{N}(\theta; 0, \Sigma) \prod_{ij} t_{ij}(\theta) \mathrm{d}\theta \simeq \prod_{ij} C_{ij} \int \mathcal{N}(\theta; 0, \Sigma) \prod_{ij} q_{ij}(\theta) \mathrm{d}\theta$$

The integral on the right hand side is a Gaussian convolution and is therefore also Gaussian. The $C_{ij}$s can be approximated by matching the zeroth order moment in the site update. As noted in the paper we can also compute the derivatives with respect to some prior hyper-parameter (see Seeger [2005]).

### 3.4  Expectation-Propagation (spike and slab prior)

The posterior can be written as

$$\pi(\theta|\mathcal{D}) \propto \prod_{i,j} t_{ij}(\theta) \prod_{k=1}^{d} t_k(\theta_k, z_k)\mathcal{B}er(z_k; p),$$

where $z_k \in \{0, 1\}$ codes the origin of $\theta_k$, spike/slab, and where $t_k(\theta_k, z_k) \propto z_k\mathcal{N}(\theta_k; 0, v_0) + (1 - z_k)\mathcal{N}(\theta_k; 0, v_1)$. The approximation given by EP is of the form,

$$Q(\theta, z) \propto \prod_{i,j} q_{ij}(\theta) \prod_{k=1}^{d} q_k(\theta_k, z_k)\mathcal{B}er(z_k; p_k),$$

where $q_k(\theta_k, z_k) \propto \mathcal{B}er(z_k, p_k)\mathcal{N}(\theta_k; m_k, \sigma_k^2)$, and $t_{ij}(\theta)$ is as in the previous section. The cavity moments are easy to compute as the approximation is Gaussian in $\theta$ and Bernoulli in $z$. In both cases we can deduce cavity moments because division is stable inside those classes of functions. We get some distribution $Q^{\backslash k}(\theta_k) \propto \mathcal{B}er(z_k; p^{\backslash k})\mathcal{N}(\theta_k; m^{\backslash k}, \sigma^{2,\backslash k})$. We can compute the normalizing constant of the distribution $Q^{\backslash ij}(\theta)t_k(\theta_k, z_k)$, namely,

$$Z_k = p^{\backslash k} \int \mathcal{N}(\theta_k; 0, v_0)\mathcal{N}(\theta_k; m^{\backslash k}, \sigma^{2,\backslash k})\mathrm{d}\theta_k + (1 - p^{\backslash k}) \int \mathcal{N}(\theta_k; 0, v_0)\mathcal{N}(\theta_k; m^{\backslash k}, \sigma^{2,\backslash k})\mathrm{d}\theta_k$$

Where we can find the update by computing the derivatives of $\log Z_k$ with respect to $p^{\backslash k}$, $m^{\backslash k}$ and $\sigma^{2,\backslash k}$

Initialization for the Gaussian is done to a given $\Sigma_0$ that will be subtracted later on. The initial $p_k$s are taken such that the approximation equals the prior $p$ at the first iteration.

## 5  Numerical illustration

Figure 1 shows the posterior marginals as given by EP and tempering SMC. The later is exact in the sense that the only error stems from Monte Carlo; we see that the mode is well approximated however the variance is slightly underestimated.

In Table 1 we show the CPU times in seconds, on all dataset studied. Experiments where run with a i7-3720QM CPU @ 2.60GHz intel processor with 6144 KB cache. Our linear model is overall faster on those datasets. A caveat is that Rankboost is implemented in Matlab, while our implementation is in C.

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

Figure 1: Comparison of the output of the two algorithms

| (a) 1st covariate | (b) 2nd covariate | (c) 3rd covariate |
|---|---|---|

| (d) 4th covariate | (e) 5th covariate | (f) 6th covariate |
|---|---|---|

(g) 7th covariate

Comparison of the Gaussian approximation obtained by Fractional EP (green) with the true density generated by SMC (blue) on the Pima indians dataset

| Dataset | Covariates | Balance | EP-AUC | GPEP-AUC | Rankboost |
|---|---|---|---|---|---|
| Pima | 7 | 34% | **0.06** | 7.75 | 3.26 |
| Credit | 60 | 28% | **1.98** | 7.59 | 56.54 |
| DNA | 180 | 22% | **11.26** | 63.47 | 141.60 |
| SPECTF | 22 | 50% | **0.25** | 63.47 | 3.55 |
| Colon | 2000 | 40% | 636.63 | **60.99** | 156.85 |
| Glass | 10 | 1% | **0.23** | 1.33 | 2.36 |

Table 1: Computation times in seconds