[Reviews · NeurIPS 2014]

Submitted by Assigned_Reviewer_13

[Summary]
The paper derives PAC-Bayesian bounds of the area under curve for two types of prior: a Gaussian prior and a spike-and-slab prior. Based on the theoretical results, the paper proposes two methods to construct a scoring function for binary classification problems based on sequential Monte Carlo and an expectation-propagation (EP) algorithms, respectively. The performance was evaluated using well-known benchmark data sets and more realistic DNA data set.

[Originality]
Practical implementation of PAC-Bayesian framework using EP algorithms has been quite limited so far; thus, the paper has good originality.

[Significance]
Theoretically sound algorithms for binary classification of unbalanced data sets are highly demanded in many fields such as bioinformatics and medical engineering. In this sense, the proposed algorithm integrated with PAC-Bayesian theory has good significance.

[Clarity]
Although there is so much information about complicated mathematics and it might be hard for general readers to understand, the important points were well summarized and there is no problem about the clarity.

[Quality]
The quality is good enough to be published as an academic paper if more information about numerical experiments is supplied.

[Other comments]
It would be more helpful if actual computation time is provided for each data set.

I (probably many readers as well wish more information about the feature selection for the DNA data set, the final illustration of Sec. 5. For example, I wonder if the selected features have some significance in genomics; how strongly correlated between variables it is, and so on.

Below is a list of some errata I found.
* Page 4, Line 1: Should "Theorem 2.1" be "Theorem 2.3"?
* Caption of Fig 1. The association between the methods (EP and SMC) and the colors (blue & green) seems the reverse.
* Table 1: The entry of (Glass, GPEP-AUC) should be in the unit of percent as the other entries do.

Summary: The paper presents new and practical algorithms integrated with PAC-Bayes theory, which potentially has a major impact on a subset of the NIPS community.

Submitted by Assigned_Reviewer_43

--- Summary of the paper
This paper establishes PAC-Bayesian bounds with AUC risk, correspond to two different prior for linear classifiers.
It also offers SMC and EP algorithms (the latter is recommended because of its light computational cost) designed for obtaining the pseudo-posterior distribution of classifier.

-- Quality
I did not follow all of mathematics in supplemental file, but the conditions, lemmas, and the derived bounds seem legitimate.
Claims of the paper are partly supported by experiments, but the presentation of the experimental results should be improved. Description of the datasets and explanation of experimental results are not very precise.
Also, somewhat sloppy mathematical expressions slightly impairs the quality of the paper.

-- Clarity
The clarity of the paper should be improved. Some mathematical symbols are used without definition, and some notations lack consistency. For example, "S" in theorem 2.3 and 2.4, and "\Xi" in subsection 3.1.

The main contributions I think of this paper are establishing PAC-Bayesian bounds with two different priors, and implementing EP algorithm for PAC-Bayesian approach.
Since description of experimental setting and discussion on the results are not enough, I think Section 4 can be removed, just mentioning that the results can be extended to non-linear case, to make room for further explanation and discussion on the results.
For example, it is better to include the number of samples of each dataset to Table 1. Also, I think the "Glass" dataset has more than two classes. I'd like to know how the authors cast the dataset to binary-labelled dataset.
Labels on axes of plots in all of the figures should be larger. They are so small that it is barely readable.
From Figure 3(a), I cannot observe the absorbing effects claimed in the paper. Also, there should be more explanation on Figure 3(b).

-- Originality and significance
I have never read papers deriving PAC-Bayesian bound for AUC risk, and hence the use of EP for estimating the Gibbs posterior with AUC risk is new to me.
The derivation of bounds and algorithm seems sound, and they agree with previously known results.
Since the proposed algorithm works well, particularly for highly unbalanced problems, the paper conveys
not only theoretically novel finding but also provides a practically useful algorithm.
Summary: This paper establishes PAC-Bayesian bounds with AUC risk, correspond to two different prior for linear classifiers.
The theorems seem valid, but experiments should be presented in better way.

Submitted by Assigned_Reviewer_45

*Summary

This paper develops theory and algorithms of a pac bayesian approach for scoring and classification with the AUC criterion.
The paper considers first a linear score function and 2 priors Gaussian and spike- and slabe (for the sparse case), and considers some mild assumptions in deriving PAC Bayesian non asymptotic bounds showing the consistency of this approach.
2 algorithms are provided to implement this approach : SMC(Sequential Monte Carlo) and EP (Expectation propagation) . Finally the paper discusses extension to non linear scores using Gaussian processes priors. Experiments are shown on UCI datasets.

*Quality ,Clarity:

This paper is clear and very well written and of a high technical quality. I did not have time to check all the proofs but they seem sound.

- While the PAC bayesian approach is interesting to explore. It is not clear how does it compare to a linear model and energy based models with thikhonov regularization for instance, this could be compared to the gaussian prior, (and L_1 regularization compared to the spike and slab prior) experimentally what is the advantage of your method in that case. It seems that the bayesian approach will need finer tuning and longer time to converge.

- Do you think you method extends to Multi-class AUC that is fairly less explored?
Would there be an advantage of the PAC Bayesian AUC for muli-class as you can have stronger priors for scoring functions that would allow better generalization in low sample regimes?

Summary: The paper has solid theoretical results and algorithmic implementations of the proposed methods.
Author Feedback
Author rebuttal: We are very grateful to the three reviewers for their helpful and supportive comments. We appreciate that two referees found the paper very clear, despite its technical nature, and we will make our best to improve further its clarity, in particular by taking into account the comments related to presentation (typos and so on). We reply below to some specific points. We also plan to write an extended version that we will submit to an academic journal.

Reviewer 13
===========

1. Actual CPU time: We are happy to include the CPU time in the revision. (A caveat is that Rankboost is implemented in Matlab, while our implementation is in C.) Quick summary: for linear scores, EP-AUC is between 10 and 50 times faster than Rankboost (except for colon dataset); for non-parametric scores (i.e. using a Gaussian process prior), it is sometimes faster, sometimes slower.

2. DNA data set: more information on this dataset may be found at:
http://www.inside-r.org/packages/cran/mlbench/docs/DNA

Reviewer 43
===========

1. Glass dataset: class 6 ("silicon") against all the other classes. This will be mentioned in the revision.

Reviewer 45
===========

1. Linear model and energy based models with Thikhonov regularization: since our focus is on AUC classification, we decided to concentrate for now on classification algorithms (such as RankBoost) specifically designed to maximise this criterion. But we keep in mind this suggestion for the extended version.

2. Multi-class AUC: We believe that extending both our results and our algorithms to the multi-class case is feasible, by following e.g. the approach of Hand and Till (2001); i.e. one assigns a score function $s_i$ to each class $i$, and take as a criterion,
$$MAUC = \sum_{i\neq j} AUC(i,j)$$
the sum, over all pairs (i,j) of classes, of $AUC(i,j)$, the AUC criterion for class i, versus class j, based on score $s_i$. Of course, the EP algorithm would have many more sites to incorporate in this case, but that seems doable if the number of classes remains small. We will mention this possible extension in the conclusion, and might explore it further in the extended version. It would also be interesting to explore stronger (hierarchical?) priors for such settings,but we have not given much thought about it for the moment.

References
==========

Hand, D. & R. Till (2001). A simple generalisation of the area under the ROC curve for multiple class classification problems. Machine Learning, 45(2):171–186, 2001.